# Biomarkers of Oxidative Stress in Healthy Infants within the First Three Days after Birth

**DOI:** 10.3390/antiox12061249

**Published:** 2023-06-09

**Authors:** Mónica Cavia-Saiz, Juan Arnaez, Amaia Cilla, Laura Puente, Laura C. Garcia-Miralles, Pilar Muñiz

**Affiliations:** 1Department of Biotechnology and Food Science, Faculty of Sciences, Universidad de Burgos, Plaza Misael Bañuelos, 09001 Burgos, Spain; monicacs@ubu.es; 2Neonatal Unit, Department of Pediatrics, University Hospital of Burgos, Islas Baleares s/n, 09006 Burgos, Spain; 3Neonatal Neurology, NeNe Foundation, 28010 Madrid, Spain; 4Department of Pediatrics, Hospital Universitario de Burgos, Islas Baleares s/n, 09006 Burgos, Spain; acilla@saludcastillayleon.es (A.C.); lpuente@saludcastillayleon.es (L.P.); lcgarciam@saludcastillayleon.es (L.C.G.-M.)

**Keywords:** oxidative stress, infant, birth, cord blood, glutathione, malondialdehyde, carbonyl, creatinine

## Abstract

The clinical relevance of stress biomarkers in newborns is well established. Currently, oxidative stress (OS) parameters are seen to play an important role in neonatal resuscitation guidelines, and a link has been observed between the amount of oxygen delivered and the level of OS and the development of various pathologies. The aim of the current study was to investigate changes in neonatal plasma and urine OS status during the first hours after birth. A lower antioxidant capacity (TAC) and higher levels of malondialdehyde in blood were observed in newborns at the time of birth compared with results 48 h postnatally. The urine revealed a significant and progressive increase in TAC and creatinine during the first 36 h of life, with a progressive decline thereafter. Meanwhile, malondialdehyde in urine samples showed no significant differences over time. Overall, the correlation between blood and urine parameters was poor, except for the relationship between umbilical vein glutathione reduced/oxidized ratio and urine malondialdehyde (r = 0.7; *p* = 0.004) and between TAC in the umbilical artery and urine (r = −0.547; *p* = 0.013). The biomarkers evaluated in this study could be established as reference values for neonatal OS.

## 1. Introduction

The physiological mechanisms of many diseases during the perinatal period are based on the oxidative damage produced by free radical accumulation. Perinatal oxidative stress concerns not only short-term complications but is also involved in the fetal programming of adult diseases [1].

The transition from the fetal to the neonatal period is characterized by metabolic and physiological changes that are accompanied by the overloading of aerobic metabolism, including the rapid passage of a relatively hypoxic intrauterine environment to an extrauterine one in which the oxygen pressure increases almost five-fold. As a result, the pro-oxidant state can cause oxidative stress (OS) in newborns [2,3]. Several studies have analyzed these processes, describing a higher degree of OS in term and especially in premature babies due to increased immaturity of antioxidant systems (both enzymatic and non-enzymatic), as well as the presence of free radicals [4,5,6]. Moreover, various events, including hypoxia, hyperoxia, and inflammation, can lead to possible multiple-organ damage, represented by the term “free radical disease–FRD”, in which oxidative damage plays a major role measurable through several OS biomarkers [3]. An increase in OS biomarkers has been observed in several neonatal conditions, and the great challenge ahead is to implement their use in clinical routines to improve the quality of care of neonatal patients [1,6,7].

The physiological trends and levels of OS biomarkers in the newborn are not well described in the literature owing to the significant ethical, operational and logistical challenges associated with obtaining blood samples from this population [8]. In most studies, the statistical power of the results could have been affected by the small sample size, and many studies focus on determinations in umbilical cord blood [9,10]. In addition, most studies show premature results with a particular pathology [11,12,13].

In view of the above, the aim of this study was to investigate the plasma and urinary parameters of OS in healthy newborns during the first days of life so that these determinations could be used as a reference to verify the physiological mechanisms involved in diseases with perinatal OS and as biomarkers, and in so doing, improve the diagnosis of perinatal diseases.

## 2. Materials and Methods

### 2.1. Selection of Newborns

The study was approved by the local research ethics committee (CEIC 1206) as part of an extensive prospective project that included research into the role of OS biomarkers in the neonatal period. Informed consent was obtained from the parents.

Our study is an observational cohort study using data prospectively entered into a database from all consecutive liveborn infants with gestational age at birth greater than 36 weeks and delivered in a tertiary university hospital during a 6-month period. In order to obtain the results of a cohort of 360 healthy newborns with limited potential risk factors for OS, infants with any of the following conditions were excluded from the study: (1) maternal chorioamnionitis, (2) signs of perinatal asphyxia at birth (intubation or cardiac massage, Apgar at 5 min ≤ 7, and/or cord arterial pH ≤ 7.00, (3) need for supplementary oxygen at birth, (4) admission for any comorbidity before hospital discharge including major congenital anomalies, complex heart disease, neurological conditions, respiratory or metabolic diseases, infection, and/or hyperbilirubinemia.

Obstetric and perinatal variables were retrieved from medical records.

### 2.2. Blood and Urine Sampling

Immediately after delivery, a segment of the umbilical cord was isolated between cord clamps. At the time of the study, the policy of our maternity unit was to perform early cord clamping and to take paired umbilical cord blood samples in all babies at the time of delivery. The umbilical artery (UA) and umbilical vein (UV) were serially punctured with a 14-gauge needle, and the blood was collected into a heparinized tube and centrifuged at 800× *g* for 10 min. We validated our cord blood gas values to obviate: (1) inadvertently sampling from the same vessel twice or (2) transposing the vessels either when taking the samples, on introduction into the analyzer, or when entering data. Therefore, we excluded all cases in which only one sample was taken and those in which the values were non-physiological (same pH value between UA and UV, or higher value of the UA with respect to the UV). Plasma aliquots were kept frozen at −80 °C. Another capillary blood sample was obtained before discharge at around 48 h of life in the well-baby nursery, together with standard metabolic screening tests. Urine samples were collected at two time points: the first was as soon as possible after birth, and the second was as close to the second blood collection as possible. Urine samples were stored frozen at −80 °C until biochemical analyses were conducted. Urine samples obtained beyond seven days of age were discarded.

### 2.3. Total Antioxidant Capacity by Ferric Reducing Ability of Plasma Method

The antioxidant capacity (TAC) in plasma and urine was assessed by its ability to reduce Fe (III) to Fe (II) (FRAP method). Briefly, 5 μL of the sample was incubated with 195 μL of FRAP reagent (sodium acetate buffer 0.3 M, pH 3.6, 20 mM FeCl_3_ and 10 mM 2,4,6-tris-2-piridil-s-triazine (TPTZ)) for 30 min at 37 °C. A blue-colored Fe (II)-TPTZ complex is formed, which absorbs light at 593 nm. Aqueous solutions of FeSO_4_ at different concentrations were prepared for calibration. The samples were analyzed in triplicate, and the results were finally expressed as micromolar of Fe (II) equivalents (μM Fe (II)Eq).

### 2.4. Glutathione Reduced/Oxidized Ratio Analysis

Glutathione reduced (GSH) and oxidized (GSSG) levels were determined in blood using the reaction between the sulfhydryl group of the GSH and DTNB (5,5′-dithio-bis-2-nitrobenzoic acid, Ellman’s reagent). The GSTNB (mixture of GSH and TNB) formed is reduced by glutathione reductase to recycle GSH and produce more TNB. The rate of TNB produced is directly proportional to the recycling reaction and is directly proportional to the concentration of GSH in the sample. The kinetics of the enzyme followed up at 410 nm at 2 min intervals for 20 min provides an accurate estimate of the total GSH content (reduced + oxidized) in the sample. Briefly, 10 μL of the sample (diluted 1/30) was neutralized with triethanolamine 4 M. Then, 10 μL of the mixture were incubated with 190 μL of the master mix, which contained potassium phosphate buffer 0.1 M, pH 7 with EDTA 1 mM, 0.3 mM NAPDH, 6 mM DTNB, and glutathione reductase. The total GSH levels were expressed per gram of hemoglobin (µmol/gHb). Quantification of GSSG was performed following GSH derivatization with 2-vinylpyridine for 1 h, and GSH was estimated by subtracting GSSG from the total GSH. The results were finally expressed as the GSH/GSSG ratio. A calibration curve was constructed by plotting the i-slope of each standard GSSG and total GSH. The samples were analyzed in triplicate, and the results were finally expressed as the GSH/GSSG ratio.

### 2.5. Malondialdehyde Analysis

Malondialdehyde (MDA) was measured in plasma and urine by HPLC according to Grotto et al. [14]. Samples of plasma and urine (75 µL) were mixed with 25 µL of water milli-Q and with 25 µL of 3 M NaOH and incubated at 60 °C for 30 min in a shaking water bath system. After this, 125 µL of 6% H_3_PO_4_ and 125 µL of 0.8% (thiobarbituric acid, TBA) were added, and the mixture was incubated at 90 °C for 45 min. The mixture was then cooled, and 50 µL of 10% sodium dodecyl sulfate (SDS) was added, and extraction with 300 µL of n-butanol was carried out by vortex-mix for 1 min followed by centrifuge at 3000× *g* for 10 min. The determinations were carried out using analytical reversed-phase HPLC on an Agilent 1100 series HPLC system (Agilent Technologies Inc., Palo Alto, CA, USA) coupled to a diode array detector. A Sherisorb^®^ ODS2 reversed phase C18 column (0.46 mm × 25 mm, 5 µm particle size; Waters Cromatografia S.A., Barcelona, Spain) was used. The chromatographic conditions were as follows: flow 1 mL/min; injection volume, 20 μL; mobile phase methanol: 50 mM potassium monobasic phosphate at pH 6.8 (40:60, *v*/*v*). The eluent was monitored at 532 nm. Peak identification was performed by comparison of retention times and diode array spectral characteristics with standard TMP (1,1,3,3-tetramethoxypropane). The samples were evaluated in triplicate, and the results were expressed as μM MDA equivalents.

### 2.6. Carbonyl Group Analysis

Protein oxidation was measured in plasma by an estimation of carbonyl groups formed using the protocol described by Levine et al. [15]. Briefly, a solution of 10 mM 2,4-dinitrophenylhydrazine (2,4-DNPH) in 10% trifluoroacetic acid was added to the 10 μL of plasma. This mixture was incubated for 1 h at room temperature, and proteins were precipitated with 500 μL of 20 % (*w*/*v*) of trichloroacetic acid, washed three times with ethanol/ethyl acetate (1:1 *v*/*v*), and then samples were centrifuged at 6000× *g* for 3 min. Finally, 1 mL of 6 M guanidine with a pH of 2.3 was added, and the samples were incubated in a 37 °C water bath for 30 min. Protein concentration was calculated by absorption spectrophotometry at 373 nm, using a molar absorption coefficient of 22,000 M^−1^cm^−1^. The samples were evaluated in triplicate, and the results are expressed as nmol/mg protein. Total protein concentration was determined with the Lowry method [16], using bovine albumin as a standard.

### 2.7. Creatinine Measurement

The urine creatinine measurement (mg/dL) was made with kinetic colorimetric Jaffe’s method, using an Auto Analyzer Hitachi Modular DPP (Roche).

### 2.8. Statistical Analysis

Numbers with percentages are presented for categorical variables and as the mean ± standard deviation (SD) and median (interquartile range) for continuous variables. Statistical analysis was performed using Stata (version 15.0) and SPSS (version 25.0) software packages. A multilevel model was used, after adjusting for individual and time, to examine the differences in the values of the OS parameters determined at two different times (at birth and at around 48 h). One-way analysis of the variance (ANOVA), using Tukey’s test, was used to determine the time-dependent change. A minimum significance level of 95% (*p* < 0.05) was set. Pearson correlation coefficient (r) was used to evaluate the correlation between the studied parameters. For correlations, urine samples collected within the first 8 h of life and those between 48+/−4 h were used.

## 3. Results

The final cohort comprised 360 newborns. The algorithm chart is shown in Figure 1, and Table 1 shows the general characteristics of mothers and their infants at birth.

Table 2 shows neonatal OS biomarkers of plasma samples from UV, UA, and 48 h after delivery, providing information on the general OS status. Levels of TAC were found to be significantly lower at birth than at 48 h after delivery. Statistical analysis showed a significant difference in the mean between UA and UV cord blood samples with 1.06 ± 0.33 mM Fe(II) Eq. in UV vs. 1.23 ± 0.45 mM Fe(II)Eq. in UA. A decrease in the levels of GSH (*p* < 0.05) was observed in blood 48 h after birth compared with umbilical cord blood, but there were no significant GSH/GSSG changes. In addition, a significant decrease in plasma MDA levels (*p* < 0.05) was observed at 48 h compared with levels in umbilical samples. No significant differences between UV and UA were observed.

OS urine markers are shown in Figure 2 and Appendix A. The urine revealed a significant and progressive increase in TAC during the first 36 h of life, with a progressive decline thereafter. Similar results were observed in creatinine levels; the maximum value for the creatinine was recorded at 36–48 h of life. In contrast, the biomarkers of lipid peroxidation levels—MDA—in urine samples that were studied showed no significant differences over time.

Overall, the correlation between blood and urine parameters was poor, except for the relationship between UV GSH/GSSG and urine MDA (r = 0.7; *p* = 0.004) and between TAC in AU and urine (r = −0.547; *p* = 0.013) (Appendix A).

## 4. Discussion

The current study analyzed the changes over time of biomarkers of OS in the first 72 h in newborns, which will influence postnatal pathologies. In addition, the correlation between OS parameters in plasma and urine was explored.

The increase in oxygen pressure at the time of birth results in the increased formation of reactive oxygen species (ROS). It is well documented that different perinatal circumstances and term delivery are associated with alterations of oxygenation involved in the pathogenesis of ‘oxygen radical disease in neonatology’, and the description of oxidative stress biomarkers has been clinically relevant [7,17]. The results of OS biomarkers have been crucial in supporting changes in the guidelines for newborn resuscitation [18]. In fact, oxygen supplementation during the stabilization of preterm infants is related to the level of plasma and/or urinary biomarkers of OS and to the development of bronchopulmonary dysplasia or lung infection [6,19]. However, the interest in OS in the newborn is not only for short-term complications; it is known that OS is involved in the fetal programming of adult diseases. Therefore, it is necessary to establish levels of OS biomarkers in healthy newborns—something which, at the present time, remains largely unknown. Because physiological changes occur continuously in the newborn, establishing reference values is difficult [8]. In order to define such reference values, it is important to have access to samples from a controlled and representative population. It is also important that the method used be non-invasive, such as cord blood and urine [18]. Therefore, in this study, we included blood samples collected from cord blood of UV and UA, blood samples 48 h after delivery and urine samples. Furthermore, our cohort of 360 neonates is larger than those of other studies.

The quantification of OS in our study is based on the measurement of biomarkers of antioxidant state (total antioxidant capacity-FRAP, GSHt and GSH/GSSG) and biomarkers of oxidative stress (malondialdehyde as products of lipid peroxidation and carbonyl groups formed from protein oxidation) [3,20,21,22]. Furthermore, the biomarkers of antioxidant capacity, lipid, and protein oxidation have been demonstrated to show specificity for OS diseases in newborns [18]. Although this technique has low sensitivity and specificity, it is easily applicable to clinical practice.

We first focused on the time of birth because, during the transition to extrauterine life, there is an increase in ROS production and a potential OS in the samples. Therefore, the biomarkers were evaluated in cord blood from arteries and veins, which represent important sources of OS biomarkers that can help to identify early high-risk in newborns [17]. Antioxidant defense in newborns matures during the late weeks of gestation, but there is also the transfer of several antioxidants across the umbilical cord. Thus, umbilical cord blood OS markers reflect maternal and neonatal conditions at the time of delivery [23]. The function of the umbilical vein is the transport of oxygenated and nutrient-rich blood to the fetal circulation, and the umbilical arteries carry deoxygenated and nutrient-depleted blood from fetal circulation. We observed lower values of TAC (FRAP) in UV than in UA at the time of delivery. This means that the fetus transfers more antioxidants to the mother than the mother transfers to the fetus, which could help the newborn to counteract physiological OS induced by pregnancy and delivery. A study by Moustafa et al. was consistent with our results, demonstrating the low TAC in umbilical venous blood by its consumption in the neutralization of the high plasma peroxide content present [24]. These results demonstrate the potential clinical utility of cord artery blood in revealing the antioxidant capacity of the fetus. Studies by Proetti and colleagues have also shown that neonatal OS is dependent on umbilical arterial oxygen levels, suggesting the potential clinical utility of OS biomarkers as a measure of fetal redox status [25].

The other antioxidant biomarker evaluated was GSH levels. GSH is the most important intracellular antioxidant produced in erythrocytes. GSH exerts its antioxidant properties by oxidation of GSH to GSSG, and the ratio GSH/GSSG has been considered an index of the redox status and, therefore, as a useful marker of diseases. No changes were observed in GSH/GSSG rate and GSH levels between UA and UV indicating that there is no change in the redox state in the newborn. These results are consistent with a compensatory response to the hyperoxic environment at birth. Therefore, TAC and GSH/GSSG values could be reference values for optimal antioxidant defense in newborns. Further studies are needed to determine changes in this value in newborn disorders.

The relationship between biomarkers of OS, MDA, and GC of the mother and the neonates has been established, showing that a high OS in the mother also induces high OS in UV [26]. In our study, in healthy newborns, we found no differences in MDA levels and GC between UA and UV of cord blood. These results show that there is a balance in OS between the mother and the newborn, possibly due to the absence of an increase in OS at the time of birth. In this respect, it is known that changes in the source of OS in the delivery, such as an increase in the lipid peroxides from the placenta that are secreted into the maternal blood, induce a peroxidation cascade [27,28]. Oxidatively damaged proteins, if not removed by proteases, can accumulate and be involved in newborn diseases. The product of lipid peroxidation MDA and carbonyl groups can exert potent biological effects and potentially mediate some of the adverse effects of oxidant injury because they are frequently used to define OS [3]. The MDA results are in line with other studies of this marker and could be construed as showing increased OS associated with delivery [29], and carbonyl group concentration is a good measure of OS because of their relatively early formation and their stability [1]. Although several studies have found different levels of protein or lipid peroxidation products in UV and UA, most of these studies evaluated the levels of plasma hydroperoxides formed in the first phase of lipid peroxidation, while MDA is one of the final products of the decomposition of hydroperoxides. Moreover, it is known that MDA is unable to cross from maternal circulation to fetal circulation [30].

Regarding the redox state at 48 h after birth resulting in slightly increased TAC, a decreased total GSH and OS biomarkers MDA levels and no changes in the GSH/GSSG rate compared with cord blood (UA, UV) are seen. These results are consistent with the finding that OS increases during pregnancy and that lipid peroxidation and protein damage are greater at the time of delivery and then decrease [31,32]. These results seem to indicate that the term fetus can respond to OS with an increase in its antioxidant capacity, as indicated by the fact that it maintains the erythrocyte GSH/GSSG in UA and UV during labor. Furthermore, the lack of change in GSSG levels indicates modulation of the redox system at birth. The measurement of GSH/GSSG could be a useful biomarker for newborns: their ratio has been considered an index of the redox status.

In addition to the previous blood studies, we performed urine studies, as in newborns, the volume of samples obtained is very limited. In addition, ethical considerations require limiting painful procedures in newborns. Therefore, OS studies in urine analysis during the first 72 h of life have been considered a suitable alternative to invasive blood-based procedures. In this study, we showed the urinary levels of TAC, MDA, and creatinine in newborns stratified by different collection time points. The results obtained for TAC-FRAP showed higher concentrations at 24–36 h than those of the first 12 h, with a gradual decline over the next few hours of life. A similar trend was observed in urine creatinine levels. The creatinine levels of the mother and fetus are similar during pregnancy, with free exchange through the placenta. It is known that during the first days of life, blood serum creatinine reflects maternal creatinine. Our results for creatinine in urine showed a peak at 36–48 h, which could be due to passive reabsorption across immature renal tubules that occurs in the first days of life [33,34]. In addition, in those first days, there are more intense changes, both in the hemodynamic and intravascular space, as well as elevation of bilirubin, which would produce greater variability in creatinine levels [35]. Further research is necessary to improve our understanding of the kinetics of renal function and biomarkers of OS in the evolution of neonates.

In regard to the biomarker of oxidative damage MDA, the study did not show any statistical significance among intervals in the 72-h period studied. Other authors observed that the levels of urine lipid peroxidation, measured as isoprostanes in the preterm infant, vary greatly, with lower values in the first 24 h and a significant increase in the first week correlating with chronic diseases [19]. Therefore, the stability of MDA levels in the first 72 h of life observed in this study could yield reference values in the healthy newborn, and the changes could be indicators of OS disorders in the first days of life.

In regard to the blood–urine correlation study, we found a positive relationship between UV GSH/GSSG and urine MDA and a negative one between TAC in UA and urine; in general, the correlation between blood and urine parameters was poor. These data suggest that urinalysis may not provide a non-invasive means of monitoring changes in blood oxidant status.

This study has some limitations. Although n is large, it was not possible to obtain results for all the parameters at the two planned times in all the infants, which is a limitation for some analyses, such as the blood–urine correlation. In addition, no parameters were determined beyond the first days of life, despite the fact that some of the parameters exert their buffer function beyond 48–72 h post-event. On the other hand, although the exclusion criteria sought to exclude neonates with conditions that could alter their redox state determined at birth and at 48 h of life, we cannot rule out the existence of some potential pro-oxidant maternal conditions.

In summary, the biomarkers evaluated in this study could be established as reference values for OS in newborns, as they are stable biomarkers of oxidative status often correlated with OS-related disorders.

## Figures and Tables

**Figure 1 antioxidants-12-01249-f001:**
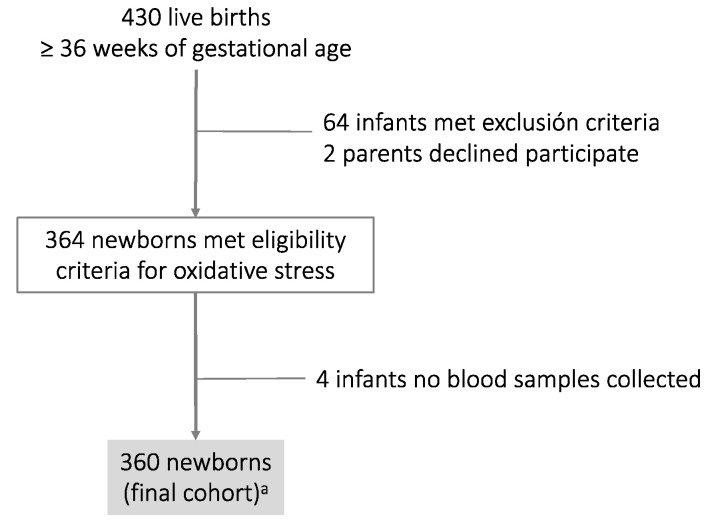
Algorithm chart of the cohort study: ^a^ Thirty-seven cord blood samples were discarded as only one sample was available (either the umbilical artery or umbilical vein). The values were non-physiological (same pH value between the umbilical artery and umbilical vein, or else the value of the umbilical artery was higher with respect to the umbilical vein).

**Figure 2 antioxidants-12-01249-f002:**
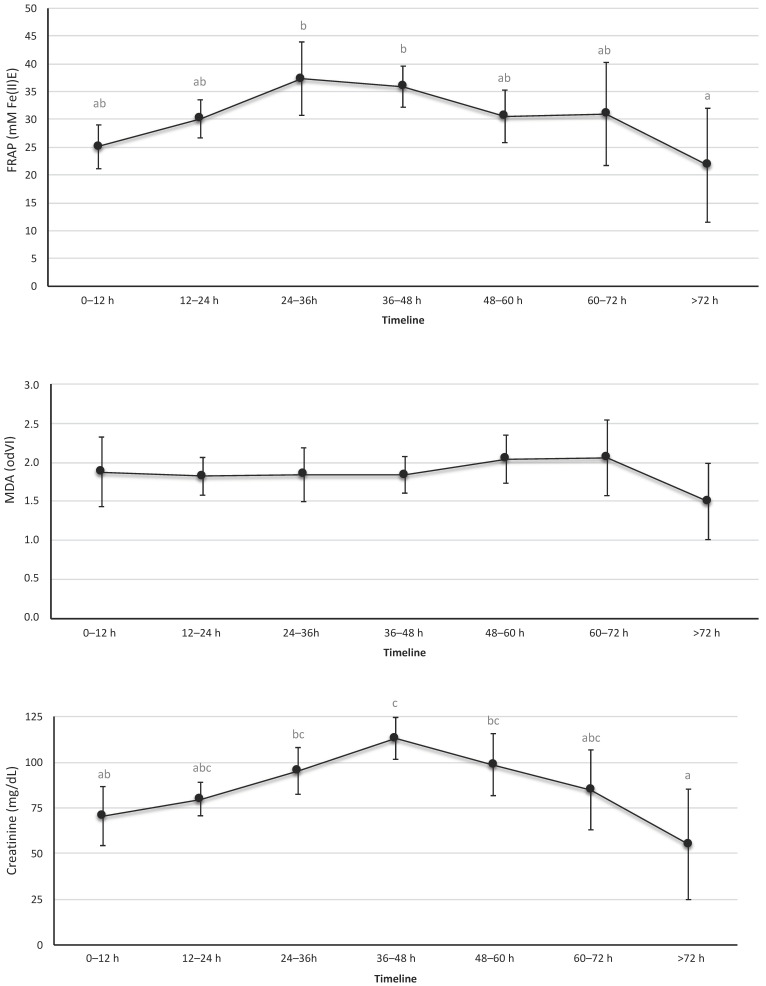
Levels of total antioxidant capacity, creatinine, and malondialdehyde in urine. These biomarkers were determined in urine samples collected at the indicated time intervals. Data expressed as mean (95% confidence interval). The letters refer to the statistical comparison using the Tukey test between the different intervals of hours of life. If an age interval shows a different letter from other intervals, it means that the difference between these two intervals is statistically significant (*p* < 0.05). In the FRAP figure, the values were different between the interval 24–36 h vs. >72 h and between 36–48 h vs. > 72 h. In the MDA figure, no interval of hours of life was found to be different from the rest. In the creatinine figure, the intervals showed a statistically significant difference between 0–12 h vs. 36–48 h, 24–36 h vs. >72 h, and 36–48 h vs. >72 h.

**Table 1 antioxidants-12-01249-t001:** General characteristics of mothers and their infants.

Maternal age (years)	33.0 (4.9)
Paternal age (years) ^a^	35.4 (5.3)
Multiple pregnancy	17/360 (5)
Preeclampsia	4/360 (1)
Gestational diabetes	29/360 (8)
Smoker during pregnancy	47/357 (13)
Hypothyroidism	40/360 (11)
Type of birth	
Eutocic	225/360 (63)
Instrumental	73/360 (20)
Elective cesarean	23/360 (6)
Urgent cesarean	39/360 (11)
Cephalic presentation	341/360 (95)
Anesthesia during labor and delivery	
No anesthesia	24/355 (7)
Local	63/355 (18)
Epidural or spinal	266/355 (74)
General	2/355 (1)
Rupture of membranes, hours ^b^	7.1 (8.4)
Stained amniotic fluid	72/360 (20)
Gestational age, weeks	39.4 (1.2)
Male	187/360 (52)
Birth weight, gr	3261 (422)
Birth length, cm	50.1 (1.9)
1-min Apgar score ^c^	8.8 (0.6)
5-min Apgar score ^c^	9.8 (0.4)
Arterial umbilical cord pH ^d^	7.22 (0.07)
Venous umbilical cord pH ^d^	7.30 (0.07)

Quantitative variables are expressed as mean (standard deviation), and categorical values are expressed as n/N (%). Data were available in 325 ^a^, 355 ^b^, 357 ^c^, and 323 ^d^ infants.

**Table 2 antioxidants-12-01249-t002:** Levels of biomarkers of oxidative stress in plasma.

Biomarker	Umbilical Vein	Umbilical Artery	Capillary Blood 48 h after Delivery	*p*-Value
FRAP (mM Fe(II)E)				
N	302	298	229	
Mean ± SD	1.06 ± 0.33	1.23 ± 0.45	1.43 ± 0.40	<0.001 ^a^
Median (IQR)	1.00 (0.83–1.21)	1.11 (0.95–1.39)	1.40 (1.18–1.66)	
GSH (μmol/gHb)				
N	262	252	187	
Mean ± SD	6.95 ± 2.94	7.04 ± 2.88	6.35 ± 2.50	0.002 ^b^
Median (IQR)	6.74 (4.89–8.55)	7.04 (4.88–8.73)	6.39 (4.83–7.96)	
GSH/GSSG				
N	272	259	207	
Mean ± SD	6.78 ± 3.35	6.46 ± 3.31	6.49 ± 3.61	0.190
Median (IQR)	6.15 (4.12–8.98)	5.72 (4.07–8.22)	5.62 (3.67–8.04)	
MDA (μM)				
N	219	204	98	
Mean ± SD	3.39 ± 1.18	3.43 ± 1.09	3.12 ± 0.96	0.035 ^b^
Median (IQR)	3.42 (2.69–4.01)	3.37 (2.77–4.05)	3.07 (2.43–3.82)	
Carbonyl Groups (nmol/mg protein)				
N	226	219	168	
Mean ± SD	1.03 ± 0.57	1.08 ± 0.69	0.99 ± 0.62	0.445
Median (IQR)	0.94 (0.62–1.33)	0.99 (0.55–1.41)	0.88 (0.49–1.42)	

CB, capillary blood; FRAP, Ferric reducing ability of plasma; GSH, total glutathione; GSH/GSSG, relation of reduced/oxidized glutathione; MDA, malondialdehyde; N, number of observations; UA, umbilical artery; UV, umbilical vein. Results are expressed as mean ± standard deviation (SD) and median (interquartile range—IQR) values. The *p*-values show comparisons among the umbilical vein, umbilical artery, and capillary blood at 48 h after delivery. ^a^ These differences were found between UV vs. UA, UV vs. CB, and UA vs. CB. ^b^ These differences were found between UV vs. CB and UA vs. CB.

## Data Availability

Data is contained within the article or Appendix A.

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
