# Peer review of "Biomarkers of Oxidative Stress in Healthy Infants within the First Three Days after Birth"

_antioxidants, 2023, doi:10.3390/antiox12061249_

Round 1
Reviewer 1 Report
This is an important topic and the manuscript is very well presented. The design is clear and appropriate. The results are interesting and provide a reference for healthy newborns. I therefore have relatively few comments.
Were the oxidative stress assay performed in duplicate? If so, please give the variance between the two. If not, please give the reliability/replicability performance for reference standards for each.
In Table 2 it would be helpful to indicate P values for the 'Umbilical artery' vs '48h after delivery' since the three way comparison is difficult to interpret. Adding 'capillary blood' for the 48h samples might be clearer for readers as well.
The Discussion is lacking a section on limitations.
Reviewer 2 Report
Dear authors, I have read your manuscript with interest, and although the topic and the background is thoroughly discussed, and references are up-to-date, I see a lot of shortcomings in your manuscript:
Major comments:
1. Materials and methods: In principle all methods should be described in detail, so that the reader can reproduce them easily. This is lacking for 2.3 and 2.4
2. Table 2 and table S1: In these tables results are presented as mean +/- SD, however the calculated mean - 2 SD, in most of the cases results in negative values, which clearly indicates that the values are not normaly distributed, therefore the use of percentiles is required.
3. The discussion is mainly a continuation of the introduction, and the results of the study are only marginally discussed.
Minor comments:
1. FigureS1, tableS1, and tableS2 could easily be incorporated in the main manuscript. It is unnecessary and laborious for the reader of an open access online journal to download additional material.
2. Legend table2 and figure1 "Different letters indicate significant differences": However, it is unclear to the reader, between which of the results that there is a significant difference.
3. p-values should be printed always the same way. Perhaps like (p< ...), an not sometimes as (P< ...)
4. Many paragraphs are hardly understandable. For example lines 202-204, 209-210. I would strongly recommend to get support from a native speaker.
Many paragraphs are hardly understandable. For example lines 202-204, 209-210. I would strongly recommend to get support from a native speaker.
Reviewer 3 Report
This paper evaluating biomarkers of oxidative stress in healthy infants in the 3 days after birth is well designed and presented in the manuscript.
The question is clinically relevant and interesting to individuals in the field of micronutrients in the perinatal period. The study design was rigorous and the population studied was representative of the described target population.
The biological samples (blood and urine) obtained were the correct samples to evaluate in this study.
The methods were well described and the studies selected to measure total antioxidant capacity (Ferric reducing ability, glutathione reduced / oxidized ratio analysis, malondialdehyde analysis, and carbonyl group analysis) were appropriately selected.
Statistical analysis was appropriate and well performed. The conclusions represent the results that were presented. The project was adequately powered with 360 newborns enrolled.
The major omission in the study design is not to evaluate and adjust for maternal smoking status which has been shown to impact oxidative stress in the maternal/fetal dyad.
Adjustment for this and preeclampsia / DM would significantly strengthen the paper.
Round 2
Reviewer 2 Report
Dear authors, thank you for the revision of the manuscript.
According to your suggestion, I would leave it to the editor to decide, wether to incorporate all supplementary material into the main text, or leave it as supplemental material.
Best regards,